

# Algal diversity of temperate biological soil crusts depends on land use intensity and affects phosphorus biogeochemical cycling

Karin Glaser[1], Karen Baumann[2], Peter Leinweber[2], Tatiana Mikhailyuk[3], Ulf Karsten[1]

[1] Institute for Biological Sciences, Applied Ecology and Phycology, University Rostock, Germany
[2] Faculty for Agricultural and Environmental Sciences, Soil Science, University Rostock, Germany
[3] M.H. Kholodny Institute of Botany, National Academy of Science of Ukraine, Tereschenkivska St. 2, UA-01004 Kyiv, Ukraine

*Correspondence to*: Karin Glaser (karin.glaser@uni-rostock.de)

## Abstract

Biological soil crusts (BSCs) form the most productive microbial biomass in many drylands and disturbed areas with a diverse microalgal community as key component. In temperate regions, BSCs are also common, but generally less studied, and they conduct important ecological functions, like stabilization of soil and enrichment of nutrients. Changes in land use and its intensity strongly influence biodiversity per se and it's role for ecosystem processes, particularly in regions which are densely populated like Europe. But systematic studies on land use (i.e. management intensity) gradients in temperate forests on BSCs are missing up to now. To close this gap of knowledge and enhance the understanding of management effects on BSCs, Cyanobacteria and eukaryotic microalgae as key primary producers of these communities were identified from pine and beech forests under different management regimes. Algae were identified morphologically based on enrichment cultivation and categorized in either coccal taxa, which occur typically in high diversity, or filamentous taxa, which have the potential to initiate BSC formation. In total, 52 algal species were recorded, most from the phylum Chlorophyta, followed by Streptophyta and Heterokontophyta; Cyanobacteria were much less abundant in temperate forest BSCs. The most abundant crust-initiating filamentous algae were three species of *Klebsormidium* (Streptophyta) a ubiquitous genus often associated with BSCs worldwide and a high tolerance to low pH. Increasing management intensity resulted in a higher richness of algae, especially the proportion of coccal algae rose. Furthermore, the proportion of inorganic phosphorus was positively correlated with the algal richness, indicating that higher diversity of algae results in a more closed P cycle. Thus, management of forests has an impact on the diversity of phototrophic organisms in BSCs, which in turn affects P cycling in the BSC.

key words: biological soil crusts, forest, management intensity, phosphorus, algae richness, *Klebsormidium*



## Introduction

Biological soil crusts (BSCs) occur on all continents on Earth, predominantly in arid and semi-arid habitats, but also in temperate regions (e.g., Belnap et al., 2001; Weber et al., 2016). In semiarid and arid environments, BSCs were studied, for example, in deserts of Israel and USA or in polar regions (Borchhardt et al., 2017; Flechtner et al., 1998; Kidron et al., 2010).

In temperate regions these habitats include dunes with sparse higher vegetation or disturbed areas in open sites (like former mining sites) (Fischer et al., 2010b; Langhans et al., 2009; Lukešová, 2001; Schulz et al., 2016). Although BSCs received raising interest in the past years, reports on BSCs from temperate forests are still missing up to now.

BSCs can be characterized as "ecosystem-engineers" forming water-stable aggregates that have important ecological roles in primary production, nitrogen cycling, mineralization, water retention, and stabilization of soils (Castillo-Monroy et al., 2010;

Evans and Johansen, 1999; Lewis, 2007). While the role of BSC in the C- and N-cycle is well documented, less is known about their role in P cycling. However, recent studies indicated that richness of BSC organisms is related to soil P content (Baumann et al., 2017; Schulz et al., 2016). Also, soil texture and carbon content seem to affect the BSC community (Elliott et al., 2014; Steven et al., 2013). But still, only little is known about environmental factors that shape BSC communities and how BSCs in turn affect soil characteristics.

Disturbance of BSCs due to land use has been reported to have strong negative effect on BSCs cover, which resulted in higher soil erosion and C and N loss in the top soil (Barger et al., 2006; Belnap, 2003). Studies on the effect of land use on BSCs were up to now exclusively conducted in arid regions. These studies reported, for example, a strong negative influence of intensive livestock grazing due to trampling on BSC cover with a recovery period of up to 27 years (Concostrina-Zubiri et al., 2014; Gomez et al., 2004; Williams et al., 2008). Also ploughing in Australian sand plains reduced the BSCs cover dramatically

(Daryanto et al., 2013). In contrast, there are no reports on land use effects in temperate regions or aspects of land use other than grazing on BSCs, like, for example, fertilization of grassland or arable land and silvicultural management.

Cyanobacteria and algae represent the most important phototrophic components of BSCs along with macroscopic lichens and bryophytes (Elbert et al., 2012). Eukaryotic algae are probably the least studied phototrophic component of BSCs, although these organisms are an essential component of BSCs because of their major contribution to C fixation (Weber et al., 2016).

BSC algae can be categorized in two functional groups. First, filamentous algae as major BSC forming taxa that stabilize soil particles by gluing them together due to the presence/excretion of mucilage. The filamentous forms occur usually in low diversity but produce high biomass. And second, coccoid algae which are attached to the soil particles or other algae and typically occur in higher diversity but lower biomass (Büdel et al., 2016).

Filamentous cyanobacteria, especially of the genus *Microcoleus*, are often the dominant phototrophic organisms in most BSCs

of drylands and in dunes from temperate regions (Garcia-Pichel et al., 2001; Schulz et al., 2016). They are described as important for BSC formation due to their ability to produce mucilage sheds and extracellular polymeric substances forming a network between soil particles (Gundlapally and Garcia-Pichel, 2006). In temperate regions, this key function is often taken





over by filamentous eukaryotic algae, like *Klebsormidium*, *Xanthonema* or *Zygogonium* (Fischer and Subbotina, 2014; Lukešová, 2001; Pluis, 1994).

The aim of the present study was to characterize for the first time the algal community in BSCs collected in temperate forests of different silvicultural management intensities. The data allowed conclusions on land-use effects on BSC algal community composition, which might result from differences in light regime affecting the phototrophs at these sites. In a previous study we provided for the first time a very detailed picture on the distribution of P content, P pools and P species in temperate BSCs and adhering soil (Baumann et al., 2017). Differences of algal richness in BSCs contributing to P cycling were detected, and the data indicated that BSCs are particularly involved in the transformation of inorganic P to organic P compounds and thus play a key role in the biologically driven P cycling in temperate soils. In addition, BSCs responded differently to management intensity depending on forest type (beech versus pine) (Baumann et al., 2017). While algal species richness of BSCs was considered as sum parameter, detailed information on the biodiversity is still missing.

Therefore, in the present study we investigated for the first time in detail the influence of forest management intensity on algal biodiversity in BSCs collected at the same plots as in Baumann et al. (2017), plus additional sampling sites. In addition, the effect of BSC algal biodiversity on C, N and P content, in particular on the different fractions of P was assessed. As a main result we can document that management of forests in temperate regions has an impact on the biodiversity of microalgae and cyanobacteria in BSCs, which in turn affects P cycling in this microecosystem.

**Material and Methods**

**Study site**

BSC samples were collected in June 2014 and 2015 as part of the DFG priority program 1374 Biodiversity Exploratories (Fischer et al., 2010a). Forest plots were sampled in the Schorfheide-Chorin Biosphere Reserve in Northeast Germany. It covers >129,000 hectares and the characteristic postglacial geomorphological structures result in diverse habitats with various forest types at the plots under study. The plots differed in the dominant tree species: Scots pine (*Pinus sylvestris* L.) or European beech (*Fagus sylvatica* L.). Further, samples were taken from natural, protected forests and from managed forest (age-class forest) (Schall and Ammer, 2013).

The top millimeters of soil, on which BSC had been visually detected as green cover, were collected by pressing a petri dish in the crust and removing gently with a spatula. After transportation to the lab the upper two millimeters of the crust was separated from the adhering soil underneath using a razor blade and stored dry in paper bags prior cultivation. In total, 31 BSCs were collected from 13 pine and 18 beech stands (Table 1).





### Culturing, identification and richness of algae

Solid 3N-Bolds Basal Medium (1.5% agar) with vitamins (Starr and Zeikus, 1993) was used for enrichment cultures in Petri dishes (9.5 cm diameter). Several 7–10 mm BSC pieces were cleaned with forceps to remove all roots and leaves to avoid the additional growth of fungi and bacteria and were placed on the surface of an agar plate under sterile conditions. Plates were incubated at 20°C, 30–35 µmol photons $m^{-2}$ $s^{-1}$ (Osram Lumilux Cool White lamps L36W/840) under a light/dark cycle of 16:8 h L:D. The plates were regularly inspected and colonies were identified four to six weeks after incubation using a light microscope (BX51, Olympus) with Nomarski differential interference optics and 1000x magnification. Light micrographs were taken with an Olympus UC30 camera attached to the microscope and processed with the software cellSens Entry (Olympus). For direct observation of BSC samples, pieces of crust were rewetted with tap water, put on slide and analyzed with the above mentioned microscope with a maximum 400x magnification.

Morphological identification of algae was based on Syllabus (Ettl and Gärtner, 1995) and, more recent publications on certain algae groups (Darienko et al., 2010; Kostikov et al., 2002; Mikhailyuk et al., 2015). Mucilage of algae was stained with an aqueous solution of methylene blue. Algae were identified that belong to Cyanobacteria, Chlorophyta, Streptophyta and some Heterokontophyta (Eustigmatophyceae). Diatoms were regularly observed in direct observation, but excluded from the analyses as the mentioned enrichment cultivation is not suitable for estimation of diatom diversity.

The richness, i.e. the number of algae species in each sample, was used as measurement for diversity rather than diversity indices like evenness, because enrichment cultivation does not allow a clear conclusion about the abundance of each species. Additionally, the community composition of the algae as reflected by the presence or absence of individual species was used as a second parameter for diversity estimation.

### Environmental variables

The forest plots were characterized by differences in the silvicultural management intensity. To evaluate the effect of management, the silvicultural management index (SMI) was used. This index takes into account the tree species, stand age and aboveground living and dead wood biomass (Schall and Ammer, 2013). Thus, the natural forest has a lower SMI than the used forest and a pine stand has a higher SMI than a beech stand.

To assess interactions between BSC biodiversity and environmental parameters, the richness, community composition of algae and proportion of filamentous algae was linked to the following environmental parameters: main tree species (pine or beech), silvicultural management intensity (SMI), water content and pH of the bulk soil for all 31 plots (water content and pH provided by I. Schöning, Table 1) and, further, for a subset of the samples (n=19), total C, N and P content, organic and inorganic P proportions, both for labile, moderately labile and stabile P. Latter are not shown here but were presented in detail by Baumann et al. (2017).



**Statistical analyses**

All statistical analyses were done using the statistical software R version 3.3.0 (R Development Core Team, 2009).

Analysis of Variance (ANOVA) was conducted to reveal the effect of environmental parameters on alga richness and proportion of filamentous algae; their best predictors were selected by backward elimination stepwise regression analysis based

on the BIC (Bayesian information criterion) using 'step' command in R. The correlation between environmental parameters were checked by Pearson correlation (cor and cor.test commands in R).

To reveal correlations of single environmental parameters with the community composition of algae, we applied PerManova (with adonis function in R (Anderson, 2001)) using the Bray–Curtis dissimilarity index (Bray and Curtis, 1957), including permutation test with 1000 permutations. The function "adonis" allows applying non-Euclidean distance metrics and handles

both categorical and continuous predictors.

For analysis of co-correlation of environmental factors pearson correlation was used. To test significant differences of environmental factors between tree species, unpaired, two-tailed t-test was performed.

Differences with a p-value below or equal to 0.05 were taken as significant.

**Results**

**Algae identification**

In total 52 different algae species were detected in enrichment cultures of all 31 BSC samples. *Stichococcus bacillaris* was the most ubiquitous taxon, observed in 27 out of 31 samples; followed by *Coccomyxa simplex* and *Klebsormidium cf. subtile* in 26 out or 23 samples, respectively. All other algal species were detected in less than 50% of the plots; 22 algal species were

observed exclusively at one plot (Figure 1). The total species richness at each plot ranged from three to 14 species, with the mean of eight and a standard deviation of 2.6 (complete species list is provided in the supplemental Table S1).

The phylum Chlorophyta made up 81% of all detected algae species, followed by Streptophyta (11%) and Heterokontophyta (6%) (Figure 2). Cyanobacteria were rare in these BSCs, just one species, *Microcoleus vaginatus*, was observed in only one sample.

The identified algae species were differentiated according to their organization form (Figure 3). We found five species with strong filaments (*Klebsormidium* cf. *flaccidum, K.* cf. *subtile, K.* cf. *nitens*, *Xanthonema* cf. *exile*, *Microcoleus vaginatus*) and two genera with short or easily disintegrated filaments (*Interfilum paradoxum*, *Stichoccous bacillaris*). In each BSC at least two different filamentous species were detected indicating their importance for BSC formation. Especially the genus *Klebsormidium* seemed to be highly important for BSCs in forest: in each BSC at least one of in total three observed

morphospecies was found (Supp. Table S1).




**Land use intensity and soil parameters**

The land use intensity was measured by applying the silvicultural management index (SMI), which based on stand density, tree species and stand age. The gravimetric water content of the bulk soil was correlated with the SMI; the pH was independent of the water content, SMI and the main tree species (Table 2).

Total C, N and P with a differentiation of P in its fractions were measured and presented by Baumann et al. (2017). Here, we analysed the correlation of this data with the SMI and pH of the forest plots.

The N content correlated with the C content and both were independent of the SMI and pH. Total P and the proportion of inorganic P were independent of the C and N content, as well as from pH and SMI (Table 2).

**Correlation of algae diversity with plot characteristics and nutrient content**

The richness of algal species and the proportion of filamentous algae in BSCs only correlated with SMI, water content and proportion of inorganic phosphorus (Table 3). All other tested parameters (C and N content, total P, proportion of organic P, pH, main tree species, and soil horizon) were excluded by stepwise model simplification based on the BIC and thus had no measurable effect on the algal species richness or proportion of filamentous algae. A higher SMI resulted in a higher species richness (Figure 2), especially the proportion of coccal algae was enhanced.

The community composition of the algae correlated with the main tree species and the water content. The SMI and proportion of inorganic P explained each 5% of the variance, but this was not significant (Table 3).

**Discussion**

**Species composition and abundance**

In total 52 algal species were identified in all BSCs sampled (Figure 1), which is a similar or lower richness compared to other reports on BSCs from temperate regions at open sites (Langhans et al., 2009; Schulz et al., 2016), but similar or higher compared to previous reports on algae from forest bulk soil (Khaybullina et al., 2010; Novakovskaya and Patova, 2008; Starks et al., 1981). Nevertheless, the given number is most probably an underestimation of the real algal biodiversity because our results are based on enrichment cultivation followed by morphological assignment. Enrichment cultivation typically covers

only cultivable algae, which represent only a small part of all algae in the BSCs (Langhans et al., 2009). Furthermore, it is not always possible to distinguish dormant from currently active microalgae. Nevertheless, direct observation of a BSC sample under the microscope gave at least a hint for the dominant active organisms. Using this approach we could prove that all filamentous algae were abundant and always alive in the BSC samples. The morphological identification of algae has known challenges: e.g. sibling species have similar characteristics but are genetically distant (Potter et al., 1997). To overcome these

limitations, researchers proposed to combine molecular and morphological methods, but also molecular techniques sometimes fail to detect some algae (Büdel et al., 2009; Garcia-Pichel et al., 2001).





All observed algal species are known as terrestrial taxa, most of them were also reported in BSCs (Büdel et al., 2016 and references therein; Ettl and Gärtner, 1995). Chlorophyceae were the most abundant phylum, which is typical for temperate regions (Büdel et al., 2016). In contrast, Cyanobacteria were represented by only one single species. Cyanobacteria are often reported as predominant species in BSCs in arid regions such as Israel and drylands of the USA (Garcia-Pichel et al., 2001; Kidron et al., 2010). Nevertheless, Cyanobacteria are less abundant in temperate regions (Gypser et al., 2016; Langhans et al., 2009; Pluis, 1994) and even rare in acidic soils, as in the forest plots of our study site Schorfheide-Chorin (Hoffmann et al., 2007; Lukešová, 2001; Lukešová and Hoffmann, 1996). Absence or presence in small amount in forest soil is concerning with low pH of soil extract which unfavorable for cyanobacteria (Hollerbakh & Shtina, 1969; Hoffmann, 1989). It seems that Cyanobacteria play only a minor role in forest ecosystems with consequences for the ecological traits that some Cyanobacteria species occupy. For example, the ability for nitrogen fixation in phototrophic organisms was only reported from Cyanobacteria and never observed in eukaryotic algae. In forest ecosystems litter and other decomposable biomass provides probably sufficient mineral nitrogen compounds, which might lead to the absence of nitrogen-fixing organisms in these systems in contrast to nitrogen-poor habitats such as dunes or deserts (Langhans et al., 2009; Schulz et al., 2016).

The filamentous alga *Klebsormidium* was found in nearly all BSCs of our study, whereas species with similar strong filaments (*Microcoleus* and *Xanthonema*) were only found occasionally. Filamentous algae can be regarded as key players in BSCs because of their potential as BSC-initiating organisms by building tight networks among particles (Büdel et al., 2016). Some investigated forest BSC were formed as well by moss protonema, which has filamentous nature and were determined as crust-forming organism (Weber et al. 2016). However, *Klebsormidium* seems to be the most important crust-initiating alga in forest ecosystems of Schorfheide-Chorin. This genus can tolerate a wide range of environmental factors and, thus, has a cosmopolitan distribution in numerous terrestrial and freshwater habitats (Karsten et al., 2016; Rindi et al., 2011 and references therein). Its presence in other terrestrial habitats such as natural rocks in plain and mountainous areas (Mikhailyuk et al., 2008), caves (Vinogradova and Mikhailyuk, 2009), sand dunes (Schulz et al., 2016), tree barks (Freystein et al., 2008), acidic post-mining sites (Lukešová, 2001), bases of urban walls (Rindi and Guiry, 2004) and building facades (Barberousse et al., 2006) is well documented. As many terrestrial algae, *Klebsormidium* is tolerant to light exposure during dehydration (Gray et al., 2007). This is a typical situation which BSC algae have to cope with because increase of light in the morning is often associated with dehydration (Raanan et al., 2016). A recent study in Central Europe, however, observed that *Klebsormidium* is sensitive to increasing light during cellular water loss (Pierangelini et al., 2017). The distribution of *Klebsormidium* in nearly all samples from Schorfheide-Chorin forest plots may be explained by a lower radiation and also lower evaporation of water in the forest ecosystem compared to open habitats (such as inland dunes), where besides *Klebsormidium* other filamentous algae were dominant (Langhans et al., 2009; Pluis, 1994). Also, the forest soil pH is rather acidic (min: 3.23; max: 3.86, Table 1) which supports a dominance of *Klebsormidium* (Škaloud et al., 2014). Thus, the low light availability, low water evaporation and the acidic soil reaction plausibly explain the presence and dominance of *Klebsormidium* as a potential BSC-initiating algal taxon in nearly all BSCs from Schorfheide-Chorin forest plots.



We identified three morpho-species of the genus *Klebsormidium* in our samples (Figure 2). All three morpho-species were reported from aeroterrestrial habitats in Central Europe (Glaser et al., 2017; Mikhailyuk et al., 2015). *Klebsormidium* has morphological features which can be easily recognized, but the identification down to species level is difficult due to morphological plasticity (Lokhorst, 1996). And still, in times of molecular identification, the debate on species definition in

the genus *Klebsormidium* is ongoing (Mikhailyuk et al., 2015; Rindi et al., 2017). Therefore, the definition of clades within *Klebsormidium* was and still is a helpful tool to differentiate between morpho- or genotypes on a species-like level (Rindi et al., 2011). Studies comparing clades at different localities on the one hand observed a global ubiquity, and local endemism on the other hand (Ryšánek et al., 2014). Especially the clade composition seems to differ depending on the habitat. In detail, *Klebsormidium cf. flaccidum* (B/C clade) was abundant in closed as well as in open habitats, whereas *K. cf. nitens* and *K. cf.*

*subtile* (E clade) were predominantly distributed in forest (Glaser et al., 2017; Mikhailyuk et al., 2015). In this study, we also observed in BSCs from forests more often *Klebsormidium cf. subtile* and *K. cf. nitens* than *K. cf. flaccidum*. Nevertheless, in desiccation experiments the recovery rates of these clades were similar (Donner et al., 2016, 2017). It is still an open question, which environmental factors caused the slight habitat preferences of the different clades. Additional ecophysiological experiments combining potential factors, such as light regimes, desiccation frequency and duration and pH, might in future

explain this habitat preferences of *Klebsormidium* clades.

Unicellular algae found in investigated soil crusts are represented mostly by Chlorophyta. High biodiversity of soil algae characteristic for humid habitats (genera Chlamydomonas, Chloromonas, Chlorococcum, Tetracystis) are typical for forest soils (Hoffmann 1989).

**Correlation with SMI**

The richness of algal species as well as the proportion of coccal algae were positively correlated with the silvicultural management index (SMI) that corresponded with conclusions about high algal biodiversity on disturbed or cultivated soils (Hollerbakh & Shtina, 1969; Hoffmann, 1989). The SMI reflects the main tree species and the stand density as a result of management practice. Most studies in the Biodiversity Exploratories on soil microorganisms in forests observed rather an effect of the main tree species on the community than of the SMI (Goldmann et al., 2015; Kaiser et al., 2016; Purahong et al.,

2014); just on study on litter decaying fungi and bacteria observed significant difference between natural and managed beech forests (Purahong et al., 2015). Kaiser et al. (2016) discussed that the different tree species influence the community of soil bacteria by shifting the pH in soil; the pH was described in this study as the main predictor for bacterial community composition. However, the differences in the bulk soil pH between beech and pine forest were not significant in Schorfheide-Chorin (Table 1) and the algae in BSCs were not affected by soil pH. We therefore rejected an effect of the SMI via the pH on

the algal species richness in Schorheide-Chorin.

However, SMI combines other potential factors which may affect BSC microalgae, namely water regime and light availability due to stand density and tree species. The sampled forest plots in the exploratory Schorfheide-Chorin were dominated by either beech or pine trees, both differing in their light regime: in beech forests the canopy shade changes during spring and therefore



radiation on the ground can be higher in winter and spring than in pine forests. Also, the stand density, another parameter of the SMI, could affect the light regime on the ground: higher density would result in less photosynthetic active radiation for photosynthetic active soil organisms. The radiation is often coupled with evaporation of pore water (Raanan et al., 2016) and, hence, the stand density could have an indirect effect on the BSC organisms via an altered water regime. Thus, we expect that

the SMI affected the algae diversity in BSC via lower light availability and lower evaporation rates. This is supported by the two-way analysis of water content and SMI, both of which is described as highly important for algal species richness. Nevertheless, it should be noted that the water content was measured in the bulk soil which might differ from that of BSC. For future studies on algal biodiversity it would be important to examine also available light on the ground and the water content in the BSC.

Although the SMI positively affected the algal richness, the community composition was correlated with the main tree species but not with the SMI. Broadleaf litter has a higher quality in terms of a more favorable C:N and C:P ratio compared to coniferous litter (Cleveland and Liptzin, 2007; McGroddy et al., 2004). It might be that the community in pine forest is shifted towards algal species, which can cope better with low nutrient availability. But also other, above mentioned factors (light regime and water evaporation) differ between both forest types and could thus cause the observed differences in the algal

community composition.

**Correlation with C/N/P**

BSCs have different important ecological functions, for example BSCs enhance the nutrient content in the top soil layer (Baumann et al., 2017; Evans and Johansen, 1999). To assess the relationship between biodiversity and biogeochemical cycling in BSCs, the content of total C, N and P and additionally the different P fractions (organic, inorganic, labile and stable fractions)

were correlated with algal diversity. Baumann et al. (2017) reported that the presence of BSCs leads to the enhanced content of total C, N and P and in particular the proportion of organic P. The present study shows that thereby the richness of algae was independent of the total C, N and P content. Thus, we assume that algal species are functional redundant and a low species richness can conduct the functional role of enhancing C, N and P content in BSCs. A more detailed analysis of the P fractions gave a slightly different picture: the proportion of inorganic P was correlated with the proportion of filamentous algae and

shows a tendency to a correlation with the richness of BSC algae. Soluble inorganic phosphate originates either from P-mineral weathering, desorption of mineral-bound phosphates or from mineralization of organic matter (Mackey and Paytan, 2009) and can be assimilated by organisms. Thus, a low amount of inorganic P could indicate a high take-up rate of BSC organisms and, thus, a more closed P cycle due to higher algae richness (Baumann et al., 2017).





**Conclusion**

This studies reports for the first time on the algae biodiversity in BSCs from temperate forests with different management intensity. The rather acidic forest soil supports a clear dominance of *Klebsormidium*-morphotypes as the main crust-initiating filamentous algae, while Cyanobacteria play a negligible role in temperate forests. Moss protonema is registered as crust-forming agent in forest ecosystems as well. Higher management intensity resulted in a higher richness of algae, especially the proportion of coccal taxa increased. We expect that the land-use intensity in forests affect the algae biodiversity via changes in the light regime (less light in high stand density and in pine forests), which is positively coupled with water evaporation. As described before, BSCs enhance the content of C, N and P compared to bulk soil(Baumann et al., 2017). In this study, we observed a relation between the proportion of inorganic P with the biodiversity of algae, especially a positive correlation with the proportion of filamentous algae. Thus, the BSC does not only enhance the total amount of P but its algal biodiversity affects the proportion of the inorganic P. Forthcoming studies should include other crust-associated organisms, like fungi and bacteria, to identify key players on the ecological role of BSCs in the P cycle.

*Competing interests.* The authors declare that they have no conflict of interest.

*Special issue statement.* This article is part of the special issue "Biological soil crusts and their role in biogeochemical processes and cycling"

*Acknowledgements.* The authors would like to thank Nadine Borchhardt for her help during BSC sampling. pH data were provided by Ingo Schöning, Theresa Klötzing and Marion Schrumpf (May Planck Institute for Biogeochemistry, Jena, Germany).
We thank the managers of the three Exploratories, Martin Gorke and all former managers for their work in maintaining the plot and project infrastructure; Christiane Fischer for giving support through the central office, Michael Owonibi for managing the central data base, and Markus Fischer, Eduard Linsenmair, Dominik Hessenmöller, Daniel Prati, Ingo Schöning, François Buscot, Ernst-Detlef Schulze, Wolfgang W. Weisser and the late Elisabeth Kalkofor their role in setting up the Biodiversity Exploratories project. The work has been funded by the DFG Priority Program 1374 "Infrastructure-Biodiversity-Exploratories" (subproject Crustfunction - KA899/28-1 and LE903/12-1). Fieldwork permits were issued by the responsible state environmental offices of Baden-Württemberg, Thüringen, and Brandenburg (according to § 72 BbgNatSchG). TM thanks the Alexander von Humboldt Foundation for financial support.





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



Table 1. Sample location, main tree species, silvicultural management index (SMI), soil horizon material, on which BSC was growing, water content and pH from bulk soil analyses (water content and pH were kindly provided by Ingo Schöning).

| Plot | latitude | longitude | main tree species | SMI | Soil Horizon | Water content (weight %) | pH (CaCl$_2$) |
|---|---|---|---|---|---|---|---|
| SW_01 | 52.9008473 | 13.846367 | pine | 0.351 | A | 12.1 | 3.64 |
| SW_02 | 52.9517289 | 13.7780281 | pine | 0.329 | B | 14.4 | 3.60 |
| SW_03 | 52.9207074 | 13.6430016 | pine | 0.334 | A | 11.7 | 3.47 |
| SW_04 | 52.9173466 | 13.8473114 | pine | 0.136 | A | 13.9 | 3.50 |
| SW_05 | 53.0570336 | 13.8853662 | beech | 0.211 | A | 13.9 | 3.42 |
| SW_06 | 53.0570336 | 13.8853662 | beech | 0.211 | A | 13.9 | 3.42 |
| SW_07 | 52.9074428 | 13.841688 | beech | 0.319 | A | 17.9 | 3.67 |
| SW_08 | 52.9074428 | 13.841688 | beech | 0.319 | C | 17.9 | 3.67 |
| SW_09 | 53.107348 | 13.6944189 | beech | 0.082 | C | 18.6 | 3.73 |
| SW_10 | 53.107348 | 13.6944189 | beech | 0.082 | A | 18.6 | 3.73 |
| SW_11 | 53.1917972 | 13.9303379 | beech | 0.059 | A | 20.7 | 3.38 |
| SW_12 | 53.1917972 | 13.9303379 | beech | 0.059 | A | 20.7 | 3.38 |
| SW_13 | 53.0445869 | 13.8101029 | beech | 0.017 | B | 16.4 | 3.56 |
| SW_14 | 53.0445869 | 13.8101029 | beech | 0.017 | C | 16.4 | 3.56 |
| SW_15 | 53.0910959 | 13.6378431 | pine | 0.381 | A | 9.9 | 3.70 |
| SW_16 | 53.0902937 | 13.6337038 | pine | 0.281 | A | 12.4 | 3.66 |
| SW_17 | 52.9179137 | 13.7521735 | pine | 0.276 | A | 15.8 | 3.38 |
| SW_18 | 52.9145421 | 13.7375526 | pine | 0.330 | C | 6.1 | 3.72 |
| SW_19 | 53.0765832 | 13.8639856 | pine | 0.335 | A | 8.4 | 3.57 |
| SW_20 | 53.0886055 | 13.6353842 | pine | 0.357 | A | 9.0 | 3.66 |
| SW_21 | 52.9155883 | 13.7404509 | pine | 0.218 | A | 13.0 | 3.44 |
| SW_22 | 52.8958259 | 13.852147 | pine | 0.217 | C | 13.3 | 3.47 |
| SW_23 | 52.8958259 | 13.852147 | pine | 0.217 | A | 13.3 | 3.47 |
| SW_24 | 52.9400216 | 13.7826121 | beech | 0.161 | A | 16.8 | 3.62 |
| SW_25 | 52.9400216 | 13.7826121 | beech | 0.161 | C | 16.8 | 3.62 |
| SW_26 | 52.9147688 | 13.8623651 | beech | 0.250 | C | 15.7 | 3.68 |
| SW_27 | 52.9147688 | 13.8623651 | beech | 0.250 | A | 15.7 | 3.68 |
| SW_28 | 52.9009765 | 13.9283256 | beech | 0.229 | A | 18.9 | 3.72 |
| SW_29 | 52.9009765 | 13.9283256 | beech | 0.229 | A | 18.9 | 3.72 |



| SW_30 | 53.0512659 | 13.8449954 | beech | 0.070 | A | 14.1 | 3.71 |
| SW_31 | 53.0512659 | 13.8449954 | beech | 0.070 | C | 14.1 | 3.71 |




Table 2. Significant Pearson correltation coefficients to reveal correlations between environmental factors, which might affect or be affected by the biodiversity of algae. This co-correlation analysis should support the correct interpretation of potential important factors of the biodiversity. SMI-silvicultural management index; n.s. – not significant

| | main tree species | SMI | water content | pH | C content | N content | P content |
|---|---|---|---|---|---|---|---|
| SMI | -0.6 | | | | | | |
| water content | 0.77 | -0.59 | | | | | |
| pH | n.s. | n.s. | n.s. | | | | |
| C content | n.s. | n.s. | n.s. | n.s. | | | |
| N content | n.s. | n.s. | n.s. | n.s. | 0.94 | | |
| P content | n.s. | n.s. | n.s. | n.s. | n.s. | n.s. | |
| proportion of inorganic P | n.s. | n.s. | n.s. | n.s. | n.s. | -0.78 | 0.6 |

Table 3. Effect of environmental factors on algae richness, filamentous algae proportion (both estimated by ANOVA) and community composition of algae (estimated by PerMANOVA) quantified by the percentage of explained variance. The significance level is indicated by: ***-p<0.001, **-p<0.01, *-p<0.05, °-p-<0.1, ns- not significant

| | algae richness | proportion of filamentous algae | community composition |
|---|---|---|---|
| SMI | 30.5 % ** | 37.7 % *** | 5.6 % n.s. |
| water content | 15.7 % * | 14.0 % ** | 9.6 % * |
| proportion inorganic P | 11.0 % ° | 29.1 % *** | 5.8 % n.s. |
| main tree species | 0.9 % n.s. | 0.3 % n.s. | 14.7 % *** |





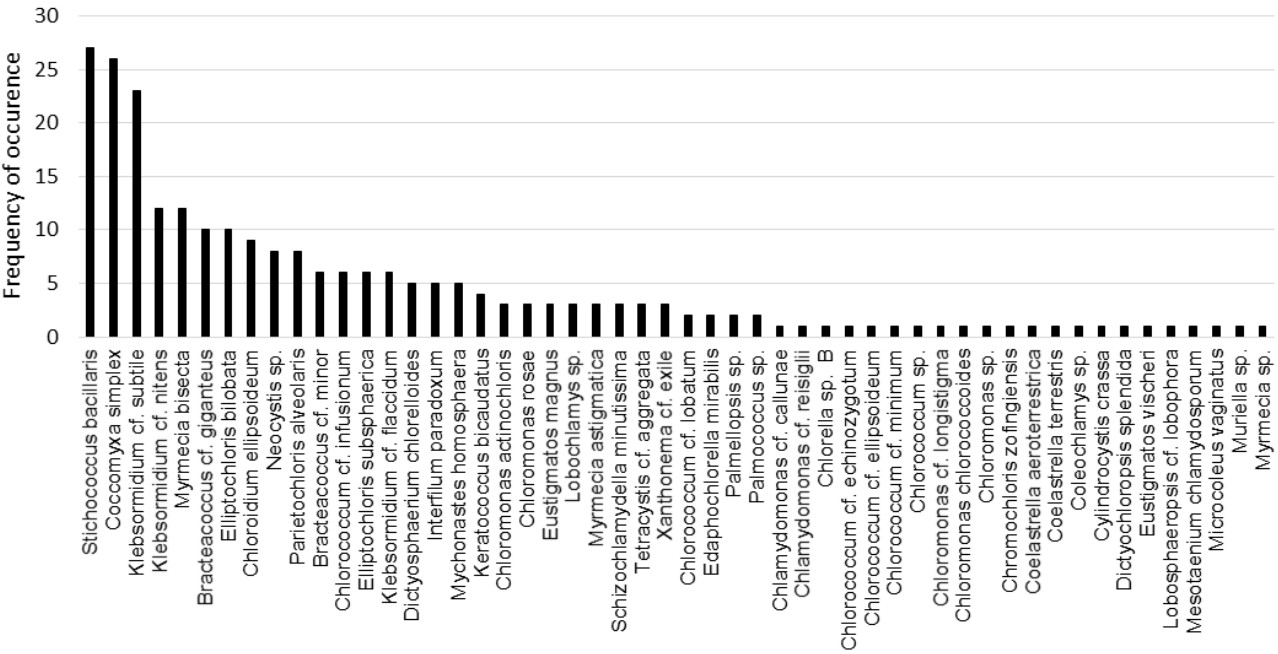

Figure 1. Frequency of occurrence of each algae species in biological soil crust from forest sites (n=31).





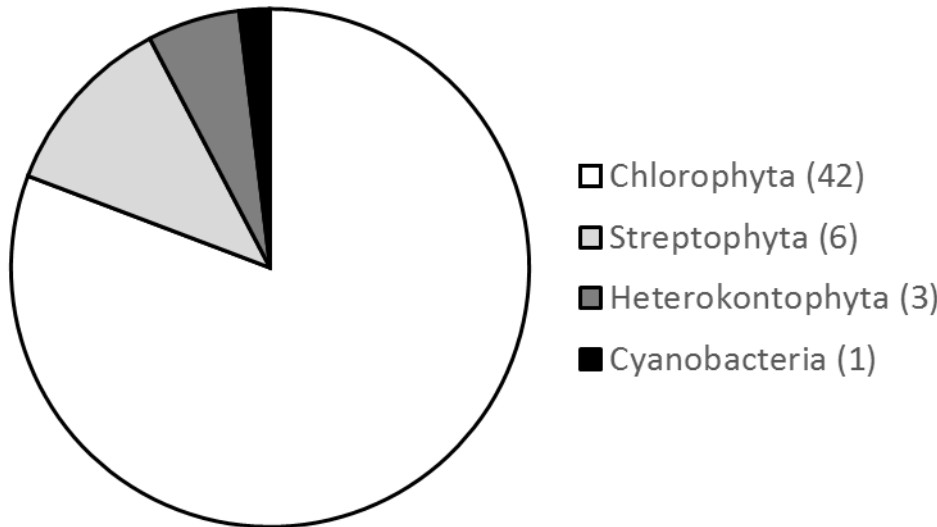

Figure 2. Distribution of algae phyla in BSCs from 31 forest plots based on enrichment cultivation and morphological identification; total number of species for each phylum is given in brackets.



Figure 3. Filamentenous and some examples of coccal algae from forest BSCs: algae with strong filaments: A-*Xanthonema cf. exile,* B-*Microcoleus vaginatus,* C-*Klebsormidium cf. flaccidum;* coccal algae: D-*Chloroidium ellipsoideum,* E-*Eustigmatos magnus,* F-*Coccomyxa simplex;* algae with short or easily disintegrated filaments*: G-Stichococcus bacillaris,* H-*Interfilum paradoxum* (mucilage bridges were coloured with methylene blue); scale bar = 5µm



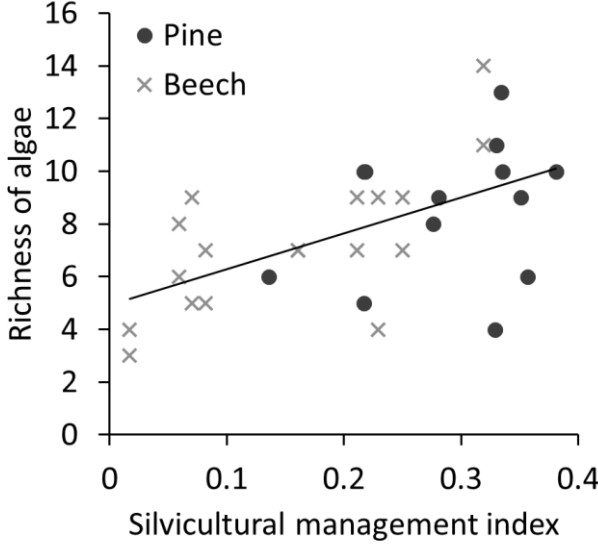

Figure 4. Plot of algae richness over the silvicultural management index (SMI), the line indicates the best linear fit (slope: 13.6, p<0.001)