# Peer review of "Algal richness of temperate biological soil crusts depends on management intensity and correlates with inorganic phosphorus"

_Biogeosciences, 2017_

## Referee Comment (RC1) · Anonymous Referee #1 · 4 Dec 2017

General Comments:

The presented paper focuses on algal diversity in biological soil crusts (BSCs) forming in temperate forests. So far little is known about the BSCs in temperate forests and what organisms create them. This makes the topic of this paper very interesting. However, the paper unfortunately does not seem to merge very deep to this topic and gives rather shallow impression with multiple inaccuracies.

As a main problem I see the way how the data for algal diversity were obtained. Even though the authors are aware of the fact that the enrichment cultivation method is not suitable for all groups of algae and cyanobacteria and that it can recover only cultivable

taxa, they still decided to use it as the only source for their data. It seems that at least part of the samples (if not all) were also observed directly in the microscope without cultivation. Why the morphological identification was not done also from these direct observations? The combination of culture dependent and independent methods would provide more accurate and detail information about the algae present in the crusts even without using the molecular methods. And the authors would be able to record not only the presence or absence of given taxa, but also their abundance. Most of the conclusions are thus limited only to the cultivable algae and not to the real forest's crust diversity.

I would appreciate if the Introduction provided more information on the BSCs in forests. Most part of the Introduction introduces BSCs as we know them from the arid regions, including their ecological roles and what threatens them there. But the desert areas and open arid sites in temperate regions are very different from temperate forests, so I think it would be useful if the authors talked a little bit more about whether these facts are true for forests as well. How are the BSCs in forests defined and established? Is "green cover" really equal to BSC? (If "just" green cover was present on a statue or a wall, it would be probably called biofilm, rather than a crust.) Why should they be interesting? Maybe providing more information about the specific sampling sites would help to clarify it as well (did the authors have to remove the litter first to look for the green cover or was the sampling done in open sites in forest, . . .). I know the sampling itself was done as part of different study, but I think this information is worth repeating (maybe as part of Table 1), because it would be important also when looking at the algal diversity and which exact factors influence it.

I am a little confused about the terms silvicultural management intensity, managed forest, and so on. Could the authors please specify more what it means in practice with regard to the BSCs? Do I understand it right that in more managed forests the soil is more often disturbed by heavy machines, traffic, etc? Maybe the authors could provide more detail on how the protected forest differed from the managed forest specifically

with regard to the crusts (overall soil cover, amount of dead biomass on the ground, density of the tree stand,...). Also it is not clear which samples were collected from the protected and which from the managed forest. The only indication the readers get is that the SMI is lower for natural forests and higher for pines. But the authors do not specify anywhere above/below what number the SMI needs to be so the forest can be considered protected/managed. Thus, in Table 1 it is not possible to find out which and how many samples were taken from which protected vs. managed type of forest (and I was not able to find it anywhere else in the text as well).

Even though the title promises information about the phosphorus biogeochemical cycling, the readers do not learn much new information and the data connected with P do not seem to be significant. The previous paper of the authors (Baumann et al., 2017) often referenced in the text seem to provide much detail information.

Specific Comments and Technical Corrections:

I would not mix algae and cyanobacteria under the name algae. Instead of "...52 different algae species..." I would consider "51 algae and a cyanobacterium" to be more precise as the prokaryotic and eukaryotic organisms are not included together.

The abbreviations "cf." are in italics in many places in the text, please check.

Methods: Study site: How many of the pinus and fagus samples originated from protected vs. managed forests?

page 2, line 3: e.g., Belnap... → e.g. Belnap (no comma)

p. 2, l. 31: mucilage SHEDS - mucilage SHEATHS maybe?

p. 3, l. 20: What does DFG stand for?

p. 3, l. 27: ...the upper two millimeters of the crust WERE...

p. 4, l. 18: Community composition based only on algae recovered by cultivation on agar plates does not reflect the real situation.

p. 5, l. 3: algaL richness

p. 5, l. 19: 26 out or 23 samples. . . confusing statement

p. 6, l. 2: , which IS based on. . .

p. 6, ls. 29-31: To overcome these limitations, researchers proposed to combine (!) molecular and morphological methods, SINCE molecular techniques ALONE sometimes ALSO fail to detect some algae.

p. 7, ls. 7-8: Absence or presence. . . The whole sentence in unclear, please check.

p. 7, l. 17: which HAS filamentous nature and WERE determined. . . unify

p. 8, l. 1: Figure 2 does not show anything about Klebsormidium morphospecies.

p. 10, l. 8: bulk soil (Baumann et al., 2017) - space missing

Table 2 : Pearson CORRELATION...

---

## Referee Comment (RC2) · Anonymous Referee #2 · 14 Dec 2017

The goal of the presented study was to describe and characterize a biological soil crust that occurs in a temperate forest in Germany (p3. L 3). This was done by assessing the species composition and evaluating the effects of the crust on the soil chemistry (C;N;P). In a second step effects of land use on the number of algal species of this crust was examined. The study is well written and might represent a new and interesting contribution. Nevertheless, there are some major drawbacks, which should be considered before publication.

1.) Definition of BSC in forests

This is a critical topic in this manuscript because the authors do not provide enough

explanations here. In the classic BSC literature, a BSC is found in " regions where water availability limits vascular plant cover" (Belnap, Weber, Büdel 2016) or "in arid and semiarid lands throughout the world, where the cover of vegetation is sparse or absent" (Belnap and Lange 2003). Both definitions are taken from the exactly cited works as given in the introduction. In the present study, the authors examine a crust from a temperate forest which contradicts this definition. A temperate forest is a habitat with a dense vegetation cover and a biome characterized by a mean precipitation between 750 and 1500 mm. This sets the presented study into a critical position for two reasons. First, because the authors do not indicate this strong discrepancy between the classical BSC definition and their own approach and explain why they still aim to refer to a BSC in this context. Secondly, because there is a vast number of publications handling effects of land use on forest understory vegetation as well as microflora and –fauna that is not considered in the discussion. It is a recent trend in BSC literature that more and more BSC are found and described in humid and forest ecosystems and I would, therefore, like to encourage to authors to critically discuss this point here, especially because this publication is part of a special issue about BSC. This study provides a chance to introduce BSC in this ecosystem if the authors try to catch up on this point and explain carefully. Statements, like given in P2l7 or P2l17 about the lack of information regarding temperate forest BSC, might just be a result from a limited literature search that focused only on BSC and not on studies regarding understory community assemblies in temperate forests. Nevertheless, as it stands now I cannot see the difference between the studies on understory forest vegetation and the presented study. In this context, the study would clearly benefit from pictures showing this crust type and how it is assembled.

2.) Diversity

According to the title, algal diversity was evaluated in this study but I wonder why this terminology was used here. Species diversity consists of three components: species richness, taxonomic or phylogenetic diversity and species evenness. With these parameters, diversity or diversity indices can be calculated. In the given study the authors provide species richness and frequency data and I think it would be more precise to refer to these (or species composition) throughout the text, rather than to diversity or even biodiversity, which wasn't studied here. The terminology should be consistent throughout the text.

(examples of the used terminology in the manuscript: diversity, biodiversity, alga richness, algal richness, community composition of alga, richness of alga, algal species richness, algal biodiversity, biodiversity of microalgae and cyanobacteria)

3.) Phosphrous biogeochemical cycling

The title promises information about phosphorus biogeochemical cycling and in the introduction, the authors state that "the effect of BSC algal biodiversity on C, N, and P content, in particular on the different fractions of P was assessed". Nonetheless, different P proportions are not shown but taken from a previous study from the authors, that is cited very often throughout the article. The only data given here are C, N, and P contents for n=19 samples which seems to be a little database for the conclusions drawn. I also wonder about the statistical test that was chosen to interpret the results, because correlation does not imply causation and the authors should be careful with interpreting their results in such a broad way.

4.) Land use intensity

Land use intensity was approximated by applying the silvicultural management index. This was determined for each study site. In table 3 it is given that SMI affects algal richness with 30,5% and the proportion of filamentous algae with 37.7%. It is stated in p 6 L 13 that higher SMI resulted in higher species richness given in Figure 2. Figure 2 represents a pie chart with mean phylum numbers of all plots. So I assume Figure 4 should show this. This graph is difficult to understand. The caption needs to be improved and explain what the symbols show. I assume these are the different forest stands and the correlation was pooled stand independently? (So why include

this information?). What was the correlation index? Where is the information about coccal algae that is given in the text? What does "richness of algae" mean? From the discussion it becomes clear that SMI basically effects the BSC composition via tree density, thus shading and light availability. Therefore, it is critical to refer to land use in this context. The authors need to define how they expect SMI to affect the BSC directly or explain the SMI was used as a proxy for tree stand density.

Additional comments: - P1l10: What do you mean with disturbed areas - P1l20: This study only examined samples from one specific area, though it's tough to generalise this finding to all temperate forests - P1l24: Please explain what mechanisms you expect to drive this relationship. Why would a higher algal species richness lead to a more closed P cycle? - P2l2: Please give exact citations on the distributions/occurrences of BSC in temperate habitats - P2l11: richness of BSC organisms? - P2l22: Elbert et al. 2012 does not distinguish between different crust components and instead pools information from all photoautotrophs in cryptogamic covers. Please find an adequate citation for your statement. - P2l24: please provide a precise citation for this statement. - P3l6: this information is irrelevant here - P3l13: which plots were these? - P3l16: what is a microecosystem? - P4l2-8: I wonder whether this cultivation technique does not influence the species assembly because some species might be excluded and others overestimated. - P4l16-18: how were the frequency data gained? How were the 'proportion' data generated? - P4l25: is richness here the total species number? - P5l3: alga richness? Did you exclude the Cyanobacteria? - P4l7: specify community composition of algae - P10l4: This statement about the moss protonema is surprising because this was not studied here and just occurred as a side note in the discussion. Why is this included in the conclusion? - P10l8: A citation of a different study in the conclusion seems misplaced. Consider summarising the data presented here. - -Table 1: this can be provided in the supplement.

Examples of literature on the effects of land use on understory forest vegetation: Marshall, V. G. (2000). Impacts of forest harvesting on biological processes in northern

forest soils. Forest Ecology and Management, 133(1), 43-60. Staddon, W. J., Duchesne, L. C., & Trevors, J. T. (1996). Conservation of forest soil microbial diversity: the impact of fire and research needs. Environmental Reviews, 4(4), 267-275. Thomas, S. C., Halpern, C. B., Falk, D. A., Liguori, D. A., & Austin, K. A. (1999). Plant diversity in managed forests: understory responses to thinning and fertilization. Ecological applications, 9(3), 864-879. Decocq, G., Aubert, M., Dupont, F., Alard, D., Saguez, R., WATTEZ‐FRANGER, A. N. N. I. E., ... & Bardat, J. (2004). Plant diversity in a managed temperate deciduous forest: understorey response to two silvicultural systems. Journal of Applied Ecology, 41(6), 1065-1079. Barbier, S., Gosselin, F., & Balandier, P. (2008). Influence of tree species on understory vegetation diversity and mechanisms involved—a critical review for temperate and boreal forests. Forest ecology and management, 254(1), 1-15.

---

## Author Comment (AC1) · 10 Jan 2018

We would like to thank the Referee for the constructive review of our manuscript. It is obvious that it was read carefully by the Referee. All arguments are solid followed by helpful advices to revise the MS.

1) The Referee sees a bigger problem with the way how the algal diversity was obtained. Our results based solely on the identification of cultivable algae. We also observed the algae directly in the crust. But in this case it is impossible to identify the algae correctly for the following reasons: although the crust were rewetted and incubated for a short time, algae are not in a good state. A lot of assimilates or irregular

shape make it very hard to see and identify the morphological characteristics. Normally in direct observations, only few cells of one species can be well observed; for correct morphological identification many cells of the same species in different states are necessary. For example, for identification of Chlorococcum-species it is necessary to observe also young cells. In a mixture like in the soil crusts, it is hard to tell if one algae is a young status of Chlorococcum, or if it belongs to some Chlamydomonas-like morphotype. This is possible in a well prepared enrichment culture, where colonies of algae are separated on the agar. Also most of the detailed morphological description in "Syllabus der Boden-, Luft- und Flechtenalgen" are based on algal cultures. It is known that environmental factors influence the morphology. Therefore, correct identification is only possible with the same or very similar approaches like in the handbooks; in this case, to use common alga media. The Referee is right, with direct observation we could have also said something about the abundance and thus about biodiversity. As we can only rely on presence/absence data, we will to change the wording throughout the text and rather use "richness" instead of "diversity" to avoid misleading implications.

2) The Referee would appreciate more information on BSC in forests. We understand the doubts of the Referee, because most literature deal with BSCs from arid regions. Thus, we will follow the suggestion to enhance the introduction part. We will also enlarge the sampling description and include some pictures from our sampling campaign, which might help to get an impression of BSCs from temperate forest.

3) The Referee would like to see the section on management intensity more specified. We understand that with more details on the silvicultural management intensity and a careful wording we can avoid confusion about it. We will also add the information about protected and used forests in Table 1 to make this point more clear.

4) The Referee sees a disagreement between the title and the conclusion of the paper. We understand the arguments of the Referee, which is in accordance with the second Referee. Of course, we don't want to make false promises. Thus, we would like to change the title "Algal richness of temperate biological soil crusts depends on

management intensity and correlates with inorganic phosphours".

5) We are very grateful for the specific comments and technical corrections, which will be all followed as suggested.

---

## Author Comment (AC2) · 10 Jan 2018

We thank the Referee for the valuable and detailed comments. It is obvious that the review took some time and the MS was read carefully. With the advices, how to improve our MS, we are motivated to revise our manuscript.

1) Definition of BSC in forest The Referee pointed out that more information is needed on forest BSCs, because it is an unusual spot. We will enlarge the introduction section for a clearer explanation of biological soil crusts in temperate forests and the differences to crusts from arid regions, where BSCs are the dominating life form. We think, that pictures from the sampling will help the readers to have a better imagination of crusts

from forest sites. The Referee proposed several publications on microflora in forest understory. We agree with the opinion that an inclusion of the mentioned publications will strengthen our discussion about the obtained results. 2 ) Diversity The Referee is right, we estimated only the richness of cultivable algae. The drawbacks of our method is described well in the MS, but we have to check the wording as the terminology might be misleading at some places.

3) Phosphorus biogeochemical cycling The Referee sees a disagreement between the title and the conclusion of the paper. We understand the arguments of the Referee, which is in accordance with the first Referee. Of course, we don't want to make false promises. Thus, we would like to change the title "Algal richness of temperate biological soil crusts depends on management intensity and correlates with inorganic phosphours". The Referee is right, a correlation does not imply causation. Thus, the interpretation of our results will be carefully checked for over-interpretation.

4) Land use intensity The Referee pointed out that the description of the silvicultural management intensity (SMI) is not sufficient at some places and more information are needed. We will explain the graph in Figure 4 more in detail and introduce the SMI more detailed, so that the reader can follow our interpretation of the results more easily.

5) Additional comments: All of this comments are helpful to improve our MS at specific points and thus all will be included in the revised version of the MS.

---

## Referee Report (RR1)

**Algal richness of temperate biological soil crusts in forests depends on management intensity and correlates with inorganic phosphorus**

**General Comments:**

I appreciate the revisions that the authors made - to me the manuscript is much more understandable and comprehensive now. The extended Introduction matches the research better. Included Figure 1 showing the sampled crusts and extended Table 1 definitely adds valuable information and help to understand the research done in this paper (and distinguish it from the research already published in Baumann et al., 2017). Changes in wording in various parts of the manuscript make the reading of the paper more fluent and the main message much clearer. However, since the inorganic P only "*showed a tendency to correlate with the richness*" I do not think this finding is bold enough to be shortened to the title to "*correlates*". Thus, the title still provides slightly misleading information and should be adjusted.

**Specific Comments:**

p.2, l.18: *Seed germination of vascular plants strongly benefits from biogeochemical activities of BSCs.* – that is not completely true even in the desert areas, seed germination of some vascular plants can be actually suppressed by the presence of BSCs

Conclusion (p.10) l.8: maybe "tree fall" instead of "wind fall"?

Table 3: Would it be possible to include the direction of the studied effects to this table? So it was clear for the readers whether for example algae richness increased with increasing water content or vice versa directly from this table? This may also help the authors with making the title of the paper more specific and corresponding more accurately with their results.

Figure 3: B – does not show any features of *M. vaginatus* (actually this piece of filament could be nearly anything), please, replace it with more illustrative picture

---

## Author Response (AR2)

General Comments:
The presented paper focuses on algal diversity in biological soil crusts (BSCs) forming in temperate forests. So far little is known about the BSCs in temperate forests and what organisms create them. This makes the topic of this paper very interesting. However, the paper unfortunately does not seem to merge very deep to this topic and gives rather shallow impression with multiple inaccuracies.

**As a main problem I see the way how the data for algal diversity were obtained. Even though the authors are aware of the fact that the enrichment cultivation method is not suitable for all groups of algae and cyanobacteria and that it can recover only cultivable taxa, they still decided to use it as the only source for their data. It seems that at least part of the samples (if not all) were also observed directly in the microscope without cultivation. Why the morphological identification was not done also from these direct observations? The combination of culture dependent and independent methods would provide more accurate and detail information about the algae present in the crusts even without using the molecular methods. And the authors would be able to record not only the presence or absence of given taxa, but also their abundance. Most of the conclusions are thus limited only to the cultivable algae and not to the real forest's crust diversity.**

*Our results based solely on the identification of cultivable algae. We also observed the algae directly in the crust. But in this case it is impossible to identify the algae correctly for the following reasons: although the crust were rewetted and incubated for a short time, not all algal species are in a reasonable morphological state because of spore formation, presence of extracellular matrix (mucilage etc.), cellular features (accumulation of storage compounds etc.) etc, all of which hamper unambiguous identification based on morphological characteristics. Normally in direct observations, only few cells of one species can be well observed; for correct morphological identification many cells of the same species in different states are necessary. For example, for identification of Chlorococcum-species it is necessary to observe also young cells. In a mixed community like in the soil crusts, it is hard to tell if one algae represents a young status of Chlorococcum, or if it belongs to some Chlamydomonas-like morphotype. This is only possible in a well prepared enrichment culture, where colonies of algae are separated on the agar.*

*Also most of the detailed morphological descriptions in "Syllabus der Boden-, Luft- und Flechtenalgen" are based on algal cultures. It is known that environmental factors influence the morphology. Therefore, correct identification is only possible with the same or very similar approaches like in algal handbooks; in this case, to use common alga media.*

*The Referee is right, with direct observation we could have also said something about the abundance and thus about biodiversity. As we can only rely on presence/absence data, we changed the wording throughout the text and rather use "richness" instead of "diversity" to avoid misleading implications. As the same misunderstanding applies for "community composition", thus, we also changed the wording to* presence or absence of individual algal *(for example, p. 4, ls. 24). As a*

second parameter, we showed similarity between single plots by presence / absence of individual species, which combines the total number and the identity of the algal species.)

5    **I would appreciate if the Introduction provided more information on the BSCs in forests. Most part of the Introduction introduces BSCs as we know them from the arid regions, including their ecological roles and what threatens them there. But the desert areas and open arid sites in temperate regions are very different from temperate forests, so I think it would be useful if the authors talked a little bit more about whether these facts are true for forests as well. How are the BSCs in forests defined and established? Is "green cover" really equal to BSC? (If "just" green cover was present on a**

10    **statue or a wall, it would be probably called biofilm, rather than a crust.) Why should they be interesting? Maybe providing more information about the specific sampling sites would help to clarify it as well (did the authors have to remove the litter first to look for the green cover or was the sampling done in open sites in forest, : : :). I know the sampling itself was done as part of different study, but I think this information is worth repeating (maybe as part of Table 1), because it would be important also when looking at the algal diversity and which exact factors influence it.**

*We understand the doubts of the Referee, because most literature deals indeed with BSCs from arid regions. Thus, we followed the suggestion to enhance the introduction part. We included some pictures from our sampling campaign, which might help to get an impression of BSCs from temperate forest (figure 1).*

20    *p. 2, ls. 7* Although BSCs received raising interest in the past years, for example, as global player in terrestrial nitrogen fixation (Elbert et al. 2012), reports on BSCs from forests are very rare (Seitz et al. 2017). Under mesic conditions BSCs have to compete with vascular plants and thus their development is often limited. Especially in forests the limitation of light and the occurrence of litter restricts the crust development. But disturbances of the higher vegetation layer change this competitive situation and allow the development of BSCs. Such disturbances occur frequently in temperate forests, for example, natural

25    tree fall, pits of wild boars, litter free spots at slopes, molehill-like humps, or human-induced disturbances such as skid trails and clear-cut areas. Especially tree falls after storm events is a rising problem in Europe due to increasing number and strengths of storms, probably because as a consequence of global change (www.dwd.de). At such spots, BSCs typically serve as pioneer vegetation for colonialization of naked soils after heavy disturbance and destruction of intact forest ecosystems. Thus, BSCs can protect disturbed areas, for example, from erosion and due to the biological introduction of carbon and nutrients into the

30    soil regrowth of vascular plants is initiated (Seitz et al., 2017). Seed germination of vascular plants strongly benefits from biogeochemical activities of BSCs (Li et al., 2005; Su et al., 2009).

*p. 3, ls. 29* Samples were taken from natural, protected forests and from managed forest (age-class forest) on disturbed areas where BSCs could develop on litter free bare soil (for illustration see Figure 1).

*p. 10, ls. 8* BSCs are able to coexist in temperate forest ecosystems, because natural and human-induced disturbances, such as

35    wind fall and skid trails, regularly provide free space for crusts to develop.

    **I am a little confused about the terms silvicultural management intensity, managed forest, and so on. Could the authors please specify more what it means in practice with regard to the BSCs? Do I understand it right that in more managed**

40    **forests the soil is more often disturbed by heavy machines, traffic, etc? Maybe the authors could provide more detail on how the protected forest differed from the managed forest specifically with regard to the crusts (overall soil cover,**

**amount of dead biomass on the ground, density of the tree stand,: : :). Also it is not clear which samples were collected from the protected and which from the managed forest. The only indication the readers get is that the SMI is lower for natural forests and higher for pines. But the authors do not specify anywhere above/below what number the SMI needs**
5 **to be so the forest can be considered protected/managed. Thus, in Table 1 it is not possible to find out which and how many samples were taken from which protected vs. managed type of forest (and I was not able to find it anywhere else in the text as well).**

*We understand that with more details on the silvicultural management intensity and a careful wording we can avoid confusion about it. We added the requested information if the sampling sites were located in a natural or managed forest in table 1 as*
10 *well as in the text.*

*p. 4, ls. 29* In natural forests, no management was conducted, meaning that fallen trees were left in place and no trees were cut. In managed age-class forests, the stands were disturbed due to e.g. usage of skid trails and removal of dead trees as well as tree cut. […] The natural forest has a lower SMI than the managed forest; a pine stand has a higher SMI than a beech stand; high stand density is reflected by a high SMI.

15 *p. 8, ls. 28* The richness of algal species as well as the proportion of coccal algae were positively correlated with the silvicultural management index (SMI), which means that we discovered more alga species in BSCs from managed than from natural forest ecosystems.

**Even though the title promises information about the phosphorus biogeochemical cycling, the readers do not learn**
20 **much new information and the data connected with P do not seem to be significant. The previous paper of the authors (Baumann et al., 2017) often referenced in the text seem to provide much detail information.**

*We understand the arguments of the Referee, which is in accordance with the second Referee. Of course, we don't want to make false promises. Thus, we changed the title to "Algal richness of temperate biological soil crusts depends on management*
25 *intensity and correlates with inorganic phosphorus".*

**Specific Comments and Technical Corrections:**
**I would not mix algae and cyanobacteria under the name algae. Instead of ": : :52**
30 **different algae species: : :" I would consider "51 algae and a cyanobacterium" to be more precise as the prokaryotic and eukaryotic organisms are not included together.**
*Good point which we addressed.*

**The abbreviations "cf." are in italics in many places in the text, please check.**
35 *Thanks for the comment, it was corrected.*

**Methods: Study site: How many of the pinus and fagus samples originated from protected vs. managed forests?**
*This information is included now in the text as well as in table 1.*
40
**page 2, line 3: e.g., Belnap: : : ! e.g. Belnap (no comma)**
**p. 2, l. 31: mucilage SHEDS - mucilage SHEATHS maybe?**
**p. 3, l. 20: What does DFG stand for?**
**p. 3, l. 27: : : :the upper two millimeters of the crust WERE: : :**
45 **p. 4, l. 18: Community composition based only on algae recovered by cultivation on**

**agar plates does not reflect the real situation.**

*We corrected the wording to presence or absence of indivdiual species.*

**p. 5, l. 3: algaL richness**

**p. 5, l. 19: 26 out or 23 samples: : : confusing statement**
**p. 6, l. 2: , which IS based on: : :**
**p. 6, ls. 29-31: To overcome these limitations, researchers proposed to combine (!) molecular and morphological methods, SINCE molecular techniques ALONE sometimes ALSO fail to detect some algae.**
**p. 7, ls. 7-8: Absence or presence: : : The whole sentence in unclear, please check.**
**p. 7, l. 17: which HAS filamentous nature and WERE determined: : : unify**
**p. 8, l. 1: Figure 2 does not show anything about Klebsormidium morphospecies.**
**p. 10, l. 8: bulk soil (Baumann et al., 2017) - space missing**
**Table 2 : Pearson CORRELATION...**
*All very specific comments were followed as suggested.*

The goal of the presented study was to describe and characterize a biological soil crust that occurs in a temperate forest in Germany (p3. L 3). This was done by assessing the species composition and evaluating the effects of the crust on the soil chemistry (C;N;P). In a second step effects of land use on the number of algal species of this crust was examined. The study is well written and might represent a new and interesting contribution. Nevertheless, there are some major drawbacks, which should be considered before publication.

**1.) Definition of BSC in forests**

**This is a critical topic in this manuscript because the authors do not provide enough explanations here. In the classic BSC literature, a BSC is found in " regions where water availability limits vascular plant cover" (Belnap, Weber, Büdel 2016) or "in arid and semiarid lands throughout the world, where the cover of vegetation is sparse or absent" (Belnap and Lange 2003). Both definitions are taken from the exactly cited works as given in the introduction. In the present study, the authors examine a crust from a temperate forest which contradicts this definition. A temperate forest is a habitat with a dense vegetation cover and a biome characterized by a mean precipitation between 750 and 1500 mm. This sets the presented study into a critical position for two reasons. First, because the authors do not indicate this strong discrepancy between the classical BSC definition and their own approach and explain why they still aim to refer to a BSC in this context. Secondly, because there is a vast number of publications handling effects of land use on forest understory vegetation as well as microflora and –fauna that is not considered in the discussion. It is a recent trend in BSC literature that more and more BSC are found and described in humid and forest ecosystems and I would, therefore, like to encourage to authors to critically discuss this point here, especially because this publication is part of a special issue about BSC. This study provides a chance to introduce BSC in this ecosystem if the authors try to catch up on this point and explain carefully. Statements, like given in P2l7 or P2l17 about the lack of information regarding temperate forest BSC, might just be a result from a limited literature search that focused only on BSC and not on studies regarding understory community assemblies in temperate forests. Nevertheless, as it stands now I cannot see the difference between the studies on understory forest vegetation and the presented study. In this context, the study would clearly benefit from pictures showing this crust type and how it is assembled.**

*We enlarged the introduction section for a clearer explanation of biological soil crusts in temperate forests and the differences to crusts from arid regions, where BSCs are the dominating life form. We included some pictures from our sampling campaign, which might help to get an impression of BSCs from temperate forest (figure 1).*

p. 2, ls. 7 Although BSCs received raising interest in the past years, for example, as global player in terrestrial nitrogen fixation (Elbert et al. 2012), reports on BSCs from forests are very rare (Seitz et al. 2017). Under mesic conditions BSCs have to compete with vascular plants and thus their development is often limited. Especially in forests the limitation of light and the occurrence of litter restricts the crust development. But disturbances of the higher vegetation layer change this competitive situation and allow the development of BSCs. Such disturbances occur frequently in temperate forests, for example, natural tree fall, pits of wild boars, litter free spots at slopes, molehill-like humps, or human-induced disturbances such as skid trails

and clear-cut areas. Especially tree falls after storm events is a rising problem in Europe due to increasing number and strengths of storms, probably because as a consequence of global change (www.dwd.de). At such spots, BSCs typically serve as pioneer vegetation for colonialization of naked soils after heavy disturbance and destruction of intact forest ecosystems. Thus, BSCs can protect disturbed areas, for example, from erosion and due to the biological introduction of carbon and nutrients into the

5    soil regrowth of vascular plants is initiated (Seitz et al., 2017). Seed germination of vascular plants strongly benefits from biogeochemical activities of BSCs (Li et al., 2005; Su et al., 2009).

*p. 3, ls. 29* Samples were taken from natural, protected forests and from managed forest (age-class forest) on disturbed areas where BSCs could develop on litter free bare soil (for illustration see Figure 1).

*p. 10, ls. 8* BSCs are able to coexist in temperate forest ecosystems, because natural and human-induced disturbances, such as

10    wind fall and skid trails, regularly provide free space for crusts to develop.

**2.) Diversity**
**According to the title, algal diversity was evaluated in this study but I wonder why this terminology was used here.**
15    **Species diversity consists of three components: species richness, taxonomic or phylogenetic diversity and species evenness. With these parameters, diversity or diversity indices can be calculated. In the given study the authors provide species richness and frequency data and I think it would be more precise to refer to these (or species composition) throughout the text, rather than to diversity or even biodiversity, which wasn0t studied here. The terminology should be consistent throughout the text. (examples of the used terminology in the manuscript: diversity, biodiversity, alga**
20    **richness, algal richness, community composition of alga, richness of alga, algal species richness, algal biodiversity, biodiversity of microalgae and cyanobacteria)**
*The Referee is right, we estimated only the richness of cultivable algae. The drawbacks of our method is described well in the*

*MS (p. 6. ls. 25). As we can only rely on presence/absence data, we changed the wording throughout the text and rather use*

*"richness" instead of "diversity" to avoid misleading implications. As the same misunderstanding applies for "community*

25    *composition", thus, we also changed the wording to* presence or absence of individual algal *(for example, p. 4, ls. 14).* As a

second parameter, we showed similarity between single plots by presence / absence of individual species, which combines the

total number and the identity of the algal species.)

30    **3.) Phosphrous biogeochemical cycling**
  **The title promises information about phosphorus biogeochemical cycling and in the introduction, the authors state that "the effect of BSC algal biodiversity on C, N, and P content, in particular on the different fractions of P was assessed". Nonetheless, different P proportions are not shown but taken from a previous study from the authors, that is cited very often throughout the article. The only data given here are C, N, and P contents for n=19 samples which**
35    **seems to be a little database for the conclusions drawn. I also wonder about the statistical test that was chosen to interpret the results, because correlation does not imply causation and the authors should be careful with interpreting their results in such a broad way.**

*We understand the arguments of the Referee, which is in accordance with the first Referee. Of course, we don't want to make*
40    *false promises. Thus, we changed the title to* "Algal richness of temperate biological soil crusts depends on management intensity and correlates with inorganic phosphorus*".*

*p. 10, ls. 14* Increasing algal richness in BSCs was supposed to enhance biogeochemical cycling of nutrients, as documented for P compared to bare soils, but this hypothesis could not be proven. Nevertheless, the fraction of inorganic P showed tendencies towards a correlation with BSC algae, especially with filamentous species. Consequently, the present study gives the first hint of a potential relation between the biogeochemical cycles in BSCs and algal species. This relation should be studied in more detail, for example, by gene expression analyses to understand if and how algae in BSCs influence the cycling of P. Also, forthcoming studies should include other crust-associated organisms, like fungi and bacteria, to identify key players on the ecological role of BSCs in the P cycle.

**4.) Land use intensity**
**Land use intensity was approximated by applying the silvicultural management index. This was determined for each study site. In table 3 it is given that SMI affects algal richness with 30,5% and the proportion of filamentous algae with 37.7%. It is stated in p 6 L 13 that higher SMI resulted in higher species richness given in Figure 2. Figure 2 represents a pie chart with mean phylum numbers of all plots. So I assume Figure 4 should show this. This graph is difficult to understand. The caption needs to be improved and explain what the symbols show. I assume these are the different forest stands and the correlation was pooled stand independently? (So why include this information?). What was the correlation index? Where is the information about coccal algae that is given in the text? What does "richness of algae" mean? From the discussion it becomes clear that SMI basically effects the BSC composition via tree density, thus shading and light availability. Therefore, it is critical to refer to land use in this context. The authors need to define how they expect SMI to affect the BSC directly or explain the SMI was used as a proxy for tree stand density.**

*We understand that with more details on the silvicultural management intensity and a careful wording we can avoid confusion about it.*

*p. 4, ls. 29* In natural forests, no management was conducted, meaning that fallen trees were left in place and no trees were cut. In managed age-class forests, the stands were disturbed due to e.g. usage of skid trails and removal of dead trees as well as tree cut. […] The natural forest has a lower SMI than the managed forest; a pine stand has a higher SMI than a beech stand; high stand density is reflected by a high SMI.

*p. 8, ls. 28* The richness of algal species as well as the proportion of coccal algae were positively correlated with the silvicultural management index (SMI), which means that we discovered more alga species in BSCs from managed than from natural forest ecosystems.

*Figure 4.* Plot of algae richness in BSCs from forests over the silvicultural management index (SMI), natural forest has a low SMI, managed forests a high SMI; the line indicates the best linear fit (slope: 13.6, p<0.001(Anova))

**Additional comments: - P1l10: What do you mean with disturbed areas -**
*p1., ls.11* disturbed areas worldwide, where higher vegetation is sparse,
**P1l20: This study only examined samples from one specific area, though it's tough to generalise this finding to all temperate forests –**
*misleading statement deleted*
**P1l24: Please explain what mechanisms you expect to drive this relationship. Why would a higher algal species richness lead to a more closed P cycle? –**
*statement deleted*

**P2l2: Please give exact citations on the distributions/occurrences of BSC in temperate habitats –**

*p. 2, ls. 5* In temperate regions these habitats include dunes with sparse higher vegetation or disturbed areas in open sites (e.g. former mining sites) (Fischer et al., 2010b; Langhans et al., 2009; Lukešová, 2001; Schulz et al., 2016).

**P2l11: richness of BSC organisms? –**

5 *p. 2, l. 30* the number of microalgae species in BSCs

**P2l22: Elbert et al. 2012 does not distinguish between different crust components and instead pools information from all photoautotrophs in cryptogamic covers. Please find an adequate citation for your statement. –**

*p. 2, l. 31* (Belnap et al., 2001)

**P2l24: please provide a precise citation for this statement.**

10 *changed to Büdel et al., 2016*

**- P3l6: this information is irrelevant here -**

*deleted*

**P3l13: which plots were these? – P3l16: what is a microecosystem? –**

*last sentence was deleted*

15 **P4l2-8: I wonder whether this cultivation technique does not influence the species assembly because some species might be excluded and others overestimated. –**

*p. 6, ls. 29* Nevertheless, the given number is most probably an underestimation of the real algal richness because our results are based on enrichment cultivation followed by morphological assignment. Enrichment cultivation typically covers mainly cultivable algae, which represent only a small part of all phototrophic microorganisms in BSCs (Langhans et al., 2009). A

20 recent paper comparing metagenomic data with morphological data based on enrichment cultivation estimated a match of about 10% of all microalgae in a polar BSC (Rippin et al., 2018). Furthermore, it is not always possible to distinguish dormant from currently active microalgae. However, direct observation of a BSC sample under the microscope gives at least a first hint for the dominant active organisms. Using this approach we could prove that all filamentous algae were abundant and always vital in the BSC samples. The morphological identification of algae has known challenges, for example, sibling species have

25 similar characteristics but are genetically distant (Potter et al., 1997). To overcome these limitations, researchers proposed to combine molecular and morphological methods, since molecular techniques alone sometimes also fail to detect some taxa based on problems with DNA extraction, appropriate primers etc. (Büdel et al., 2009; Garcia-Pichel et al., 2001).

**P4l16-18: how were the frequency data gained? How were the 'proportion' data generated? –**

30 *p. 4, ls. 25* Further, the identified algae were categorized in filamentous or coccal life form, because both differ in their ecological function. Filamentous algae, in contrast to coccal algae, have the potential to initiate crust-formation and stabilize soil particles by gluing them together.

**P4l25: is richness here the total species number? - P5l3: alga richness? Did you exclude the Cyanobacteria? –**

*p. 4, ls. 22* richness of algae (total number of algae and cyanobacteria species per sample)

35 **P4l7: specify community composition of algae –**

*p.4, ls. 23* As a second parameter for biodiversity the similarity between single plots is shown as reflected in the presence / absence of individual species which combines the total number and the identity of all algal taxa observed.

**P10l4: This statement about the moss protonema is surprising because this was not studied here and just occurred as a side note in the discussion. Why is this included in the conclusion? –**

40 *as suggested, deleted from the conclusion*

**P10l8: A citation of a different study in the conclusion seems misplaced. Consider summarising the data presented here.**

*We rewrote this paragraph accordingly.*

**Table 1: this can be provided in the supplement.**
*We decided to keep table 1 in the main text but added two information: managed or unmanaged site; proportion of inorganic P*

[revised manuscript text omitted]

15    different management intensity. The rather acidic forest soil supports a clear dominance of streptophycean *Klebsormidium*-morphotypes as the main crust-initiating filamentous algae, while Cyanobacteria always play a negligible role. in at temperate forestsour study site. Moss protonema is registered as crust forming agent in forest ecosystems as well. Higher forest management intensity resulted in a higher richness of algae, especially the proportion of coccal taxa increased. We It is expect reasonable to assume that the land usesilvicultural management intensity in forests affect the algale biodiversity richness via

20    due to, for example, higher stand density in managed forests, which changes in the light and water regime. (less light in high stand density and in pine forests), and thus,also the which is positively coupled with water evaporation. Increasingregime. Increasing algal richness in BSCs was supposed to enhance biogeochemical cycling of nutrients, as documented for P compared to bare soils, but this hypothesis could not be proven.
We expected a correlation between the total content of carbon, nitrogen and phosphorusphosphorus, N, and P with the number

25    of algal species or their identity, respectively, because it was
As described before, that BSCs enhance the content of C, N and Pnutrients compared to bulk soil(Baumann et al., 2017). In contrast , we observed no correlation between the total content of C, N and P and the species richness of algae and Cyanobacteria. Nevertheless, the fraction of inorganic phosphorusP showed tendencies towards a correlation with the BSC algae in biological soil crustsBSCs, 
[revised manuscript text omitted]

---

## Author Response (AR3)

**First Reviewer**

**Suggestions for revision or reasons for rejection (will be published if the paper is accepted for final publication)**

The revision of the manuscript improved most of the topics issued in the first discussion round. Nevertheless, there is still potential for further improvement that can be solved in a minor revision of the manuscript.

The new section in the introduction describes very clearly that BSC can occur in the forest context but only in open sites, where the continuous forest is disturbed. As a consequence, interpretations on larger scales (forest as an ecosystem, meso- or macro- scale) should be taken with great care. It should be stated throughout the manuscript that this study focuses on disturbed sites with minimal areal extension in forests (a fact that also becomes very clear from the chosen pictures in Figure 1)!

There are several occasions throughout the text where this still needs clarification (examples):

Maybe, even the title should be revised according to that new way of showing and interpreting the data.

We decided to change the title: "Algal richness in BSCs from forest under different management intensity with some implications for P cycling"

P3 line 14: collected from disturbed/open sites within temperate forests
*changed as proposed*

P10 line 9: For the first time, algal richness in BSCs from such disturbed sites in temperate forests with different management intensities were described.
*changed as proposed*

P2 line 25: It could be stated here, that in contrast to the studies in arid ecosystems, here, not the disturbance of the crusts was studied, but rather how the disturbance of the continuous forest effects the crust development (crust formation via changes in species richness).
*P2 ls 24. In contrast, there are no reports on land use effects in temperate regions or aspects of land use other than grazing or human activities on BSCs. Also missing are reports on the benefits for BSCs in terms of coverage due to disturbances in continuous vegetation like forests.*

P8 line 28: This perfectly suits your idea, of BSC occurring in disturbed sites with advantages compared to the higher vegetation. Maybe you can elaborate a bit more on this here.
*P8 ls 31. The silvicultural management index (SMI) was used to estimate the management intensity. It takes into account the tree species, stand age and aboveground living and stand density. However,*

*intensive managed forest did not necessarily inherit more disturbed sites suitable for BSC development. In contrast, high stand density (typical for intensively managed stands) reduces the amount of potential disturbed areas needed for BSC development. However, managed forests have a higher risk for complete stand loss: either because of regular clear-cut or strong storms; it is more likely to lose huge areas in pine stands with high density compared to natural beech forest.*

Other comments:
P1 line 15: Delete: but and replace with: anyhow,
*changed as proposed*

P1 line 23: delete one of the two ..
*changed as proposed*

P2 line 18: The citations of Li et al. 2005 and Su et al. 2009 do not make sense in this context because they both refer to desert plants and are therefore not representative in the forest context.
*We decided to delete the sentence about seed germination, because it is anyway a very speculative aspect and suitable citations from temperate regions are rare.*

P3 line 4: see also Szyja et al. 2018, published in this same special issue.
*Thank you for the comment. The reference was included at this position as well as earlier in the introduction (P2 line 7, P3 line 4).*

P3 line 18: delete one of the two ..
*changed as proposed*

P3 line 17: Insert a space between soils and (
*changed as proposed*

P4ls 25: My original question here was: how were the frequency data gained?
*I mentioned "frequency of occurrence" at Figure 2. I meant, in how many crusts were this algal species found out of the total number of crusts observed (31). I rephrase Figure 2, because it seemed to be confusing.*
How were the 'proportion' data generated?Could you please explain?
*P4 ls 26. The proportion of filamentous algae on total number of algae was used for statistical analyses.*

P4 ls 23: Please revise to: As a second parameter for biodiversity the similarity between single plots was shown by presence/absence data of individual species, combining the total number and the identity of all algal taxa observed.
*changed as proposed*

P5 line 4: It is not clear what the authors mean with "interactions between BSC biodiversity indicators and environmental parameters". Could this be: a linkage between BSC development and environmental parameters?
*P5 l 5. To assess potential linkages between BSC organisms and environmental parameters*

P6 line 10-23: Please include some information about the implications of these correlations.
*P6 ls 11. The correlation analyses between environmental factors were conducted to understand the interrelation between the factors, which might be a driver for algae in BSCs.*
*P6 ls 20. BSCs with higher algal richness tended to a lower proportion of inorganic P.*
*P6 ls 24. This implies an effect of the main tree species and the water content on the community composition of algae in BSCs.*

P 10 ls. 8: are able to coexist WITH forest, rather than IN
*changed as proposed*

Table 2: According to a previous comment please specify the terminology: biodiversity of algae.
Figure 1: Please explain the picture in the right lower corner. Is this also taken from a root plate? Please include a scale and the picture labelling, as also commented by the editor.
*We included the picture labelling a, b, c and d and mentioned that the one crust was taken from the root plate.*

I also strongly agree with the editor, that the manuscript needs language polishing. Mainly the lack of commas in the text makes some sentences very difficult to understand.
*We checked the text and improved the language at various positions throughout the manuscript.*

**Second Reviewer**

**General Comments:**

5  I appreciate the revisions that the authors made - to me the manuscript is much more understandable and comprehensive now. The extended Introduction matches the research better. Included Figure 1 showing the sampled crusts and extended Table 1 definitely adds valuable information and help to understand the research done in this paper (and distinguish it from the research already published in Baumann et al., 2017). Changes in wording in various parts of the manuscript make the reading of the paper more fluent and the

10  main message much clearer. However, since the inorganic P only "*showed a tendency to correlate with the richness*" I do not think this finding is bold enough to be shortened to the title to "*correlates*". Thus, the title still provides slightly misleading information and should be adjusted.

**Specific Comments:**

15  p.2, l.18: *Seed germination of vascular plants strongly benefits from biogeochemical activities of BSCs.* – that is not completely true even in the desert areas, seed germination of some vascular plants can be actually suppressed by the presence of BSCs
*Also the first reviewer had similar concerns about this sentence. Thus, we decided to delete the sentence about seed germination, because it is anyway a very speculative aspect and suitable citations from*

20  *temperate regions are rare.*

Conclusion (p.10) l.8: maybe "tree fall" instead of "wind fall"?
*changed accordingly*

25  Table 3: Would it be possible to include the direction of the studied effects to this table? So it was clear for the readers whether for example algae richness increased with increasing water content or vice versa directly from this table? This may also help the authors with making the title of the paper more specific and corresponding more accurately with their results.
*Thanks for the useful remark, we included these symbols in the table.*

30

Figure 3: B – does not show any features of *M. vaginatus* (actually this piece of filament could be nearly anything), please, replace it with more illustrative picture

*changed accordingly*

[revised manuscript text omitted]

---

## Author Response (AR4)

Dear Bettina Weber,

Thanks for your detailed remarks. We asked for help and corrected punctuation and other language issues throughout the manuscript.

I hope, our manuscript fulfills the standards of the journal now.

Kind regards,

Karin Glaser

[revised manuscript text omitted]